# Reviewing challenges in access to oral health services among the LGBTQ+ community in Indiana and Michigan: A cross-sectional, exploratory study

G. Tharp[1], Manisha Wohlford[2], Anubhuti Shukla [3] *

1 Department of Sociomedical Sciences, Columbia University, New York, NY, United States of America,
2 Indiana University School of Dentistry, Indianapolis, Indiana, United States of America, 3 Department of Cariology, Operative Dentistry and Dental Public Health, Indiana University School of Dentistry, Indianapolis, Indiana, United States of America

These authors contributed equally to this work.

* anshukla@iu.edu

**Data Availability Statement:** All relevant data are within the paper and its Supporting Information files.

## Abstract

### Objective

In healthcare settings, lesbian, gay, bisexual, transgender, and queer (LGBTQ+) populations often experience discrimination, leading to decreased healthcare services utilization. In this study we have tried to identify oral healthcare providers (OHP)'s perceptions toward LGBTQ+ patients, perceived barriers for LGBTQ+ patients in accessing oral health services, and whether they were open to inclusive oral healthcare practices. In addition, the experiences of LGBTQ+ patients in oral healthcare settings including their oral healthcare seeking behaviors and beliefs were also explored.

### Methods

Descriptive, quantitative surveys were administered to OHPs and LGBTQ+ patients within Indiana and Michigan. Surveys contained questions about participant demographics, including gender and sexual minority status, and the presence of inclusive healthcare practices within the oral healthcare settings. Descriptive analyses and regression modeling were used to explore the distribution of participant responses and to identify predictors associated with patient comfort and OHP's attitudes toward LGBTQ+ patients.

### Results

Overall, 71% of LGBTQ+ patients reported regularly attending dental appointments; however, 43% reported feeling uncomfortable going to appointments and 34% reported being treated unfairly during appointments because of sexual orientation. Among OHPs, 84% reported that the healthcare settings where they practiced were welcoming for LGBTQ+ populations and 84% reported willingness to improve LGBTQ+ care. The presence of inclusive healthcare practices predicted comfort for LGBTQ+ patients (P < 0.10). Additionally,

**Funding:** Yes. The project implementation was supported by Delta Dental Foundation of Indiana. The funders played no role in study design, data collection and analysis, decision to publish, or preparation of the manuscript. https://www.deltadentalin.com/giving-back

**Competing interests:** The authors have declared that no competing interests exist.

OHPs who either identified as an ally or as having a family member or close friend in the LGBTQ+ community had higher odds of feeling responsible to treat LGBTQ+ patients.

## Conclusion

Many LGBTQ+ patients often experience discomfort in oral healthcare settings. While OHPs were largely unaware of this, evidence suggests the need for cultural competency training for OHPs.

## Introduction

In healthcare settings, lesbian, gay, bisexual, transgender, and queer (LGBTQ+) populations often experience discriminatory environments resulting from explicit or implicit biases held by healthcare providers [1–3]. Current literature has identified a positive association between transgender patients' experience of discrimination in oral healthcare settings and their level of dental fear [4]. A study conducted by the Center for American Progress (2018) indicated that discrimination experienced by LGBTQ+ patients in healthcare settings may largely affect their ability and intent to access care [5]. Dental anxiety resulting from explicit or implicit biases in oral healthcare settings can prevent LGBTQ+ patients from seeking or following through with regular dental care appointments may lead to adverse oral health outcomes.

Existing literature reveals disparities in overall health, including oral health, among the LGBTQ+ communities [6, 7]. Lesbian, gay, and bisexual (LGB) individuals have a higher risk of poor mental health, smoking, and substance misuse than heterosexual individuals [8, 9]. Increased risk of oral disease is found in the LGBTQ+ population because of depression, side effects of medications, and harmful eating behaviors [10]. Human Papillomavirus associated oropharyngeal cancers and Human Immunodeficiency Virus infections, among other sexually transmitted diseases with oral health implications, are also more prevalent in the LGBTQ + population [11]. Limited evidence from literature indicate there may be an association between hormone replacement therapy (HRT) and periodontal disease in transgender community [12, 13]. Due to the associated stigma, many transitioning transgender patients refrain from discussing this with their oral health providers and may develop concerns as possible effects of HRT on oral health [14].

Oral health is often a mirror image of an individual's overall health; in fact, several oral diseases are associated with systemic conditions and vice versa. Periodontal disease negatively affects glycemic control; this relates to the causal relationship between poor oral health and diabetes [15, 16] Oral health outcomes affect more than just the periodontium. In addition to diabetes, poor oral health has been linked to cardiovascular and rheumatoid disease [17–19]. Oral health is also known to have a positive relationship with self-esteem [20], positive quality of life and affects presenteeism, performance at work and academic achievement [21–23].

Although there is an upward trend in the acceptance and visibility of transgender and non-conforming adolescents in seeking oral health services, a great deal of work still needs to be done to improve their patient experience while navigating the system [24] and reducing disparities in accessing oral health services. A recent study conducted across different disciplines, specifically reported that dental students had significantly less positive perception of their formal training in LGBTQ+ health and more stereotypical attitudes towards LGBTQ+ populations, when compared to medical and nursing trainees [25]. With this project, we aim to evaluate the experiences of LGBTQ+ patients in oral healthcare settings including their health

seeking behaviors alongside the beliefs, perceptions, and attitudes of oral healthcare professionals toward the LGBTQ+ community and towards implementing inclusive oral healthcare practices in their offices.

## Methods

### Data source and ethical considerations

From August 2020 to February 2021, descriptive, quantitative surveys were administered to OHPs and LGBTQ+ patients. For this study, OHPs were defined as dentists, dental hygienists, dental assistants, and administrative support staff (in dental clinics) and LGBTQ+ patients were defined as individuals who were part of these clinics' patient base and who self-identified as being either a sexual and/or gender minority.

The Indiana University School of Dentistry Institutional Review Board approved this investigation (Protocol #: 2007467343) as an exempt study because the study did not require the collection of any identifiable data from any participants. This designation waived the IRB need for formal written consent. Respondents read a statement online which informed them that the study was completely voluntary, summarized the purpose of the study, and how their data would be used. Respondents were informed that by completing the survey they were offering their consent to participate. All survey data were stored in the Qualtrics online data application (26). The authors disclose no conflicts of interests.

The participants (both OHPs and LGBTQ+ patients) were recruited from dental clinics at Indiana University School of Dentistry, invited federally qualified health centers (FQHCs) and community clinics within the states of Indiana and Michigan. Electronic survey links were shared with the oral health providers' listservs at the collaborating organizations. The survey for patients was distributed via the patient portal at these clinics and included a screening question which asked participants of their gender orientation and they could proceed with the survey only if they confirmed they were part of the LGBTQ+ community. Participation for LGBTQ+ patients were incentivized and patients who completed the survey were provided a $10 Amazon gift card. There we no incentives offered to the OHPs.

### Survey tool

Both surveys contained multiple choice questions about participant demographics, including their gender and sexual minority status. The survey administered to OHPs consisted of 29 questions and included Likert scale questions about OHP's perceptions of the LGBTQ+ community, presence of inclusive healthcare practices within their office settings and their attitudes toward implementing changes to improve patients' quality of care in terms of inclusive healthcare practices. The term "inclusive healthcare practices" referred to questions around medical history, office environment and use of preferred pronouns for the LGBTQ+ patients. The survey administered to LGBTQ+ patients was 21 questions long and included Likert scale questions on their oral healthcare seeking behaviors, perceived barriers to accessing oral healthcare, and experiences relating to accessing oral healthcare.

Both surveys are attached under Appendix I in S1 File.

### Dependent variables

The outcomes of interest for this study were LGBTQ+ patients' comfort in attending dental appointments and OHP's sense of responsibility to treat patients who are members of the LGBTQ+ community. LGBTQ+ patient comfort in attending dental appointments was measured by a single Likert scaled item asking patients whether "[they are] comfortable going

dental appointments." This item was dichotomized to compare strongly disagree/disagree/and neutral relative to responses of agree/strongly agree. OHP sense of responsibility to treat members of the LGBTQ+ community was similarly measured by a single Likert scaled item asking OHPs whether "[They] feel it is [their] responsibility to care for patients who are part of the LGBTQ+ community." This item was also dichotomized to compare strongly disagree/disagree/and neutral relative to responses of agree/strongly agree.

### Independent variables

LGBTQ+ patient perceptions regarding the presence of inclusive healthcare practices in the oral healthcare setting where they receive care were used to predict LGBTQ+ patient's comfort in attending dental appointments. Inclusive healthcare practices included the presence of posters or artwork that specifically caters to the LGBTQ+ community, the ability of patients to indicate their preferred pronouns on medical forms and being treated the same as patients who are not members of the LGBTQ+ community. To predict OHP's sense of responsibility to treat members of the LGBTQ+ community, questions related to OHP self-reported affiliations (with the LGBTQ+ community) were used.

### Statistical analysis

We used counts and percentages to describe the population characteristics of study participants and to explore the distribution of participant responses to measures of inclusive healthcare practices in oral healthcare settings. LGBTQ+ status among both participant groups were also compared descriptively. Subsequently, logistic regression models were constructed to predict LGBTQ+ patient's comfort attending dental appointments and OHP's sense of responsibility toward treating LGBTQ+ patients. Each model's design was informed by the previously described independent and dependent variables. These models were evaluated at a $P < 0.10$ significance level and did not include demographic confounders. This exclusion was made to avoid bias related to this study's use of convenience sampling and the class imbalance caused by participant self-reporting. All analyses were completed using an opensource statistical software, RStudio.

### Results

In total, 255 OHPs and 248 LGBTQ+ patients participated in this study. Of enrolled LGBTQ + patients, the median age was 29 years (IQR 25–36). This sample was demographically diverse with 89% (n = 221) of LGBTQ+ patients identifying with a non-heterosexual sexual identity and 21% (n = 52) identifying with a transgender gender identity. Among enrolled OHPs, the median age was 46 years (IQR, 33–59), with 63% (n = 145) reporting having worked in their current setting > 4 years. Within the OHP population, 52% (n = 132) were dentists, 14% (n = 36) were dental assistants, 12% (n = 31) were dental hygienists, and 20% (n = 51) were administrative staff members. Demographically, we observed that our population of OHPs was much more hegemonic with 89% (n = 206) of OHPs identifying as straight or heterosexual and 100% (n = 231) identifying as cisgender. When OHPs were asked to self-identify and define their specific affiliation with the LGBTQ+ community, we found that 28% (n = 66) had no affiliation, 52% (n = 120) had a family member or close friend who is a member of the LGBTQ+ community, 28% (n = 64) were allies to the LGBTQ+ community, and that only 5% (n = 11) self-identified as a member of the LGBTQ+ community.

## Participant characteristics

For the LQBTQ+ patients enrolled in this study, we observed high levels of oral healthcare seeking attitude with 75% (n = 187) believing that dental care is a necessity and that it affects one's overall health. This additionally translated into a moderately high level of oral healthcare maintenance with 71% (n = 168) of LGBTQ+ patients reporting to have regularly (at least twice a year) visited their dentist prior to the onset of the COVID-19 pandemic. In contrast to these levels, we found that 43% (n = 106) of LGBTQ+ patients felt uncomfortable seeking care and that 34% (n = 84) believed they had been treated unfairly in oral healthcare settings because of their sexual identity or orientation (Table 1).

## Patient oral healthcare seeking behaviors and perceptions of oral healthcare settings (Table 2)

Through logistic regression modeling we identified that, when present in oral healthcare settings, inclusive healthcare environments in oral healthcare settings were found to predict significantly higher odds of patient comfort (Model 1, P<0.10). Of all the inclusive healthcare strategies, the ability for patients to indicate their preferred pronouns on medical history

**Table 1. Participant characteristics.**

| Participant Characteristics | Oral Health Providers | | Patients | |
|---|---|---|---|---|
| | Analytic Sample | | Analytic Sample | |
| **Age** | 217 | Median (IQR) | 248 | Median (IQR) |
| | | 45 (32–58) | | 29 (24–35) |
| | | N (%) | | N (%) |
| **Sexuality** | 232 | | 248 | 27 (11) |
| Straight or heterosexual | | 206(89) | | 120(48) |
| Lesbian, gay, or homosexual | | 8 (3) | | 78(31) |
| Bisexual | | 18 (8) | | 23(9) |
| Something else/not listed | | 0 | | |
| **Gender** | 231 | | 248 | |
| Male | | 85 (37) | | 121(49) |
| Female | | 146(63) | | 75(30) |
| Genderqueer | | 0 | | 19(8) |
| Trans Female | | 0 | | 18(7) |
| Trans Male | | 0 | | 15(6) |
| **Sex Assigned at Birth** | 232 | | 246 | |
| Male | | 86(37) | | 137(56) |
| Female | | 146(63) | | 109(44) |
| **Race** | 231 | | 247 | |
| White or Caucasian | | 196 (85) | | 175(71) |
| Black or African American | | 11(5) | | 54(22) |
| Asian | | 7(3) | | 3(1) |
| American Indian/Alaskan Natives | | 1(0) | | 4(2) |
| Hawaiian/Pacific Islander | | 0 | | 0 |
| Other | | 11(5) | | 2(1) |
| Multiracial | | 5(2) | | 9(4) |
| **Ethnicity** | 202 | | 244 | |
| Hispanic | | 16(8) | | 74(30) |
| Non-Hispanic | | 186(92) | | 170(70) |

**Table 2.**

| LGBTQ+ Patient Oral Healthcare Seeking Behaviors and Perceptions of Oral Healthcare Settings | N (%) Agreement |
|---|---|
| I think dental care is a necessity and it affects my overall health and quality of life (n = 248) | 187 (75) |
| I am comfortable going to dental appointments (n = 248) | 142 (57) |
| I think that people in the LGBTQ+ community have additional obstacles accessing healthcare (n = 248) | 159 (64) |
| This dental clinic has created a welcoming space for the LGBTQ+ community (n = 244) | 108 (44) |
| In the dental clinic there are posters/ artwork that specifically caters to the LGBTQ+ community in waiting room areas/common spaces (n = 244) | 67 (27) |
| The medical history forms used in this facility have a place for me to indicate my preferred pronouns (n = 244) | 90 (37) |
| The staff in this dental clinic treats patients in the LGBTQ+ community the same as heterosexual, non-transgender (cis gender) patients (n = 241) | 105 (44) |
| It takes a long time to get a dental appointment at my clinic (n = 244) | 84 (34) |
| It takes a long time to get a dental appointment at my clinic (n = 244) | 168 (71) |
| In the last 12 months/prior to COVID-19, did you visit your dentist regularly (at least twice a year) (n = 236) | 124 (50) |
| In the last 12 months/ prior to COVID-19, did you have instances where you needed to see your dentist, but could not due to financial issues (n = 248) | 85 (34) |
| In the last 12 months/ prior to COVID-19, have you been treated unfairly at this dentist's office because of your sexual identity or sexual orientation (n = 248) | 287 (75) |

forms produced statistical significance at a higher level (P<0.05) than what was observed with other strategies.

When investigating the barriers faced by our LGBTQ+ patient population, we found that financial barriers to care affected 50% (n = 124) of our patient population. While patients also reported barriers in scheduling appointments (Table 2.), this impacted a minority of participating patients. However, we ultimately observed that the majority of LGBTQ+ patients (64%, n = 159) believed that people in the LGBTQ+ community have additional obstacles accessing healthcare.

## Model 1: Environment based predictors of patient comfort in oral healthcare settings (Table 3)

Overall, a large majority (84%, n = 191) of OHPs believed that the oral healthcare setting where they work has created a welcoming environment for members of the LGBTQ

**Table 3.**

| Independent Variable | Odds Ratio | 90% C.I. | |
|---|---|---|---|
| In the dental clinic there are posters/ artwork that specifically caters to the LGBTQ + community in waiting room areas/common spaces** | 2.8525 | 1.4790 | 5.6600 |
| The medical history forms used in this facility have a place for me to indicate my preferred pronouns* | 2.3977 | 1.1.3274. | 4.3734 |
| The staff in this dental clinic treats patients in the LGBTQ+ community the same as heterosexual, non-transgender (cis gender) patients** | 2.9180 | 1.7146. | 5.0210 |

Odds ratios (OR)s and 90% confidence intervals (CI)s calculated from logistic regression model

* P < 0.05

** P < 0.10

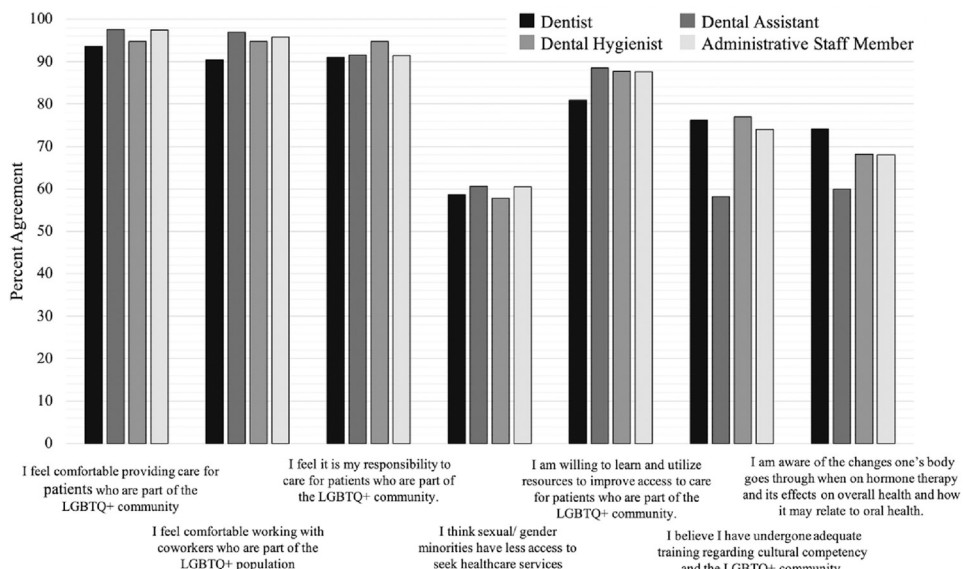

**Fig 1. Provider beliefs and attitudes toward members of the LGBTQ+ community.**

+ community and 98% (n = 223) believed that the staffs there treat patients who are members of the LGBTQ+ community the same as those who are not. More than half 54% (n = 123) reported positively when asked whether they knew who they should ask if they had a question relating to LGBTQ+ care.

A difference in beliefs, attitudes, and perceptions of the LGBTQ+ community was observed across all four OHP groups (Fig 1). While these differences were not statistically significant, we observed that, on average, dentists were less likely to agree that they were comfortable providing care to LGBTQ+ patients, were comfortable working with LGBTQ+ coworkers, reported feeling they had a responsibility to care for patients who are members of the LGBTQ+ community, and were willing to learn and utilizing resources to improve access to care for patients who are a part of the LGBTQ+ community. Further, across all provider groups, we observed low levels of agreement relating to OHPs being aware of the health disparities faced by LGBTQ+ populations, having undergone adequate training regarding cultural competencies and the LGBTQ+ community, and being aware of the impacts of HRT on an individual's overall health and specifically on oral health.

### Model 2: Affiliation-based predictors of perceived responsibility to treat LGBTQ+ patients among dentists (Table 4)

Among dentists, we explored how affiliation to the LGBTQ+ community might impact a dentists' sense of responsibility to treat members of the LGBTQ+ community. While there was no statistically significant relationship found between dentists identifying as having no affiliation or as being a member of the LGBTQ+ community and their sense of responsibility, we did find that either having a family member or close friend who is a member of the LGBTQ+ community or identifying as an ally produced a statistically significant odds (P<0.01) of feeling a responsibility to treat LGBTQ+ patients.

### Discussion

In our sample of LGBTQ+ patients and OHPs, we observed a need to improve oral healthcare practices in dental settings and to provide educational trainings for OHPs related to LGBTQ

**Table 4.**

| Independent Variable | Odds Ratio | 90% C.I. | |
|---|---|---|---|
| Dentist identified as having no affiliation with the LGBTQ+ community | 1.1142 | 0.9428 | 1.3169 |
| Dentist identified as being a member of the LGBTQ+ community | 0.9930 | 0.6894 | 1.4304 |
| Dentist identified as having a family member or close friend who is a member of the LGBTQ+ community* | 1.2546 | 1.0798 | 1.4577 |
| Dentist identified as an ally to the LGBTQ+ community** | 1.1512 | 1.0086 | 1.3138 |

Odds ratios (OR)s and 90% confidence intervals (CI)s calculated from logistic regression model

* P < 0.05

** P < 0.10

+ populations and their health. Our results were comparable to some of the previously conducted research in terms of difficulties in accessing care among the LGBTQ+ patients and the need for a holistic and welcoming attitude among health professionals [26]. This may be different from few of the other studies which report transgender and non-conforming adolescents and young adults have minimal difficulty receiving oral health care [24, 27]. Although there may not be much "tooth level" difference in treating this population, there is an obvious need for developing trusting, non-judgmental patient-doctor relationship in caring for this vulnerable group [28]. Literature indicates that knowledge and attitudes among healthcare providers about homosexuality, are influenced by them being an ally or having a friend who is from the LGBTQ+ community [29].

Increasingly over the past decade, public health literature has emphasized the role that inclusive healthcare environments play in improving health outcomes among marginalized populations (2). Related interventions often take the form of educational toolkits or continued education opportunities; however, in some instances, interventions have also been deployed at schools of medicine in the form of curricula changes [25]. Within the field of public health dentistry, there has been little to no emphasis on the development of such interventions to improve care of marginalized populations or, more specifically, LGBTQ+ patients. Our results indicated that introducing established, inclusive healthcare practices would improve the comfort of LGBTQ+ patients in oral healthcare settings. We found that, while agreement differed across provider types, all groups expressed moderately high agreement that they were willing to learn and utilize resources to improve access to care for LGBTQ+ patients. This level of agreement indicates a similarly high level of interest among OHPs to engage with LGBTQ + health education and training.

## Limitations

Our study was limited by several factors. For instance, the convenience-sampling techniques limit the representativeness of the sample. Our patient population included those that were currently receiving dental care, that does present a selection bias. Many LGBTQ+ patients, delay or completely forgo medically necessary care for fear of stigma unlike our patient sample that had a dental home. This study was a cross-sectional, exploratory study due to which, we are unable to make claims of generalizability of the study results. Additionally, we did not formally validate our survey prior to data collection, though items were reviewed by an interprofessional group of collaborators where the study was implemented. With the topic in question, social desirability bias may be another limitation, especially for the OHP's survey. Although the surveys were anonymous, many participants may not have responded honestly about

negative attitudes towards or discomfort with LGBTQ individuals. It's important to note that information on the educational status of the patient pool which is an important variable and can directly affect their health literacy and ability as well as intent to avail oral healthcare services, wasn't collected. The study sites were in relatively conservative states of Indiana and Michigan which may as well confound the study results. There are currently no comprehensive civil rights protections in place for LGBTQ individuals at the state level in Indiana [29]. Michigan has one of the highest levels of hate crimes reported per 100,000 residents and the overall environment is not very LGBTQ+ friendly [29].

## Further research and practical implications

The need to improve the health, safety, and well-being of LGBTQ+ individuals is also listed as a goal for Healthy People 2020 [30]. Poor attitude towards LGBTQ+ individuals can affect resource utilizations and LGBTQs health status implications for oral health [31].

Future research in this direction will help identify gaps and direct oral health policies to improve access to care for the LGBTQ+ community. Provider stigma attached with treating this population is a huge issue and motivating providers to engage with this population and influencing their attitudes may require a lot of efforts in terms of cultural humility and workplace inclusiveness training for the OHPs. Educators should develop training programs that provide OHPs with the knowledge and skills to ensure LGBTQ+ patients receive effective oral health care when they access services for themselves.

## Supporting information

**S1 File. This is the S1 File (Appendix).** This is the S1 Appendix I.
(DOCX)

**S2 File. This is the S2 File (Minimal dataset).** This is the S2 Data tables.
(XLSX)

## Acknowledgments

We would like to thank all staff members of Corktown Health based in Michigan; especially Teresa Roscoe, the Chief Operating Officer for helping with data collection and offering administrative support for this study. We would also like to acknowledge the efforts of Dr Karla Marin, Oral health Director -Indiana Primary Healthcare Association (IPHCA) and Indiana University School of Dentistry for their assistance in survey distribution and data collection.

## Author Contributions

**Conceptualization:** Anubhuti Shukla.

**Data curation:** Manisha Wohlford.

**Formal analysis:** G. Tharp.

**Funding acquisition:** Anubhuti Shukla.

**Investigation:** Anubhuti Shukla.

**Methodology:** G. Tharp.

**Project administration:** Anubhuti Shukla.

**Resources:** Manisha Wohlford, Anubhuti Shukla.

**Software:** G. Tharp.

**Supervision:** Anubhuti Shukla.

**Validation:** Anubhuti Shukla.

**Writing – original draft:** Anubhuti Shukla.

**Writing – review & editing:** G. Tharp, Manisha Wohlford, Anubhuti Shukla.

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
