## [Decision Letter · Decision Letter 0]

26 Jan 2022

PONE-D-21-36125Reviewing Challenges in Access to Oral Health Services Among the LGBTQ+ Community in Indiana and Michigan: A Cross-Sectional, Exploratory StudyPLOS ONE

Thank you for submitting your manuscript to PLOS ONE. After careful consideration, we feel that it has merit but does not fully meet PLOS ONE’s publication criteria as it currently stands. Therefore, we invite you to submit a revised version of the manuscript that addresses the points raised during the review process.

Please submit your revised manuscript by Mar 12 2022 11:59PM If you will need more time than this to complete your revisions, please reply to this message or contact the journal office at plosone@plos.org. Please include the following items when submitting your revised manuscript:A rebuttal letter that responds to each point raised by the academic editor and reviewer(s). You should upload this letter as a separate file labeled 'Response to Reviewers'.A marked-up copy of your manuscript that highlights changes made to the original version. You should upload this as a separate file labeled 'Revised Manuscript with Track Changes'.An unmarked version of your revised paper without tracked changes. You should upload this as a separate file labeled 'Manuscript'.

We look forward to receiving your revised manuscript.

Kind regards,

Luigi Lavorgna

Academic Editor

PLOS ONE

Journal Requirements:

“Last but not the least, we would extend our heartfelt gratitude to the Delta Dental Foundation of Indiana for recognizing the importance of this study and offering financial support for the execution of the study.”

 “Yes. The project implementation was supported by Delta Dental Foundation of Indiana. The funders played no role in study design, data collection and analysis, decision to publish, or preparation of the manuscript.

https://www.deltadentalin.com/giving-back”

Reviewers' comments:

Reviewer's Responses to Questions

**Comments to the Author**

1. Is the manuscript technically sound, and do the data support the conclusions?

Reviewer #1: Yes

2. Has the statistical analysis been performed appropriately and rigorously? 

Reviewer #1: Yes

3. Have the authors made all data underlying the findings in their manuscript fully available?

Reviewer #1: Yes

4. Is the manuscript presented in an intelligible fashion and written in standard English?

Reviewer #1: Yes

5. Review Comments to the Author

Reviewer #1: Tharp and colleagues reported on oral healthcare providers (OHP)’s perception and practice toward LGBTQ+ patients, and, conversely, the experiences of LGBTQ+ patients in oral healthcare settings. The manuscript is overall clear and well written. Methods are sufficiently sound. The topic is interesting, since policies towards LGBTQ+ community can ultimately enhance their healthcare access and health status. I only have some minor comments to the authors.

The study was conducted in Indiana and Michigan. A brief summary of the overall LGBTQ+ attitude in these states should be added to put results into different contexts.

“The survey for patients was distributed via the patient portal at these clinics”. How many participants were potentially reached? How many actually commenced on the survey? And how many screened negative? I believe this information

Was educational status collected? This is quite relevant variable, also to be considered ins statistical models.

In the discussion, authors should mention that poor attitude towards LGBTQ+ individuals can affect resource utilizations and LGBTs health status (10.1016/j.msard.2017.02.001), also with implications for oral health.

6. PLOS authors have the option to publish the peer review history of their article (what does this mean?). If published, this will include your full peer review and any attached files.

Reviewer #1: No

---

## [Author Response · Author response to Decision Letter 0]

31 Jan 2022

2/1/2022

PONE-D-21-36125

Reviewing Challenges in Access to Oral Health Services Among the LGBTQ+ Community in Indiana and Michigan: A Cross-Sectional, Exploratory Study

Dear Editors,

We would like to thank the reviewers for their valuable time and effort in reviewing our manuscript. We have addressed all the comments and made edits in the blinded manuscript suing track changes as recommended. 

The manuscript has been formatted to comply with the PLOSone style. The funding information has been removed from Acknowledgments section and updated in the funding statement. Study’s minimal underlying data set is uploaded as Supporting Information file and references checked.

Please find our responses to the comments and suggestions below, for additional clarification. 

Reviewer #1: Tharp and colleagues reported on oral healthcare providers (OHP)’s perception and practice toward LGBTQ+ patients, and, conversely, the experiences of LGBTQ+ patients in oral healthcare settings. The manuscript is overall clear and well written. Methods are sufficiently sound. The topic is interesting, since policies towards LGBTQ+ community can ultimately enhance their healthcare access and health status. I only have some minor comments to the authors.

The study was conducted in Indiana and Michigan. A brief summary of the overall LGBTQ+ attitude in these states should be added to put results into different contexts.

Response: Thanks for the excellent suggestion, we included this information under discussion section of the manuscript.

“The survey for patients was distributed via the patient portal at these clinics”. How many participants were potentially reached? How many actually commenced on the survey? And how many screened negative? I believe this information

Response: It’s not possible to get this information as each of the participating clinics because of privacy laws, but it was distributed to all the patients attending those clinics. The total number of patients who commenced was the total number of patient responses we received

Was educational status collected? This is quite relevant variable, also to be considered ins statistical models.

Response: we understand the usefulness of collecting this information but unfortunately, we did not ask for this in our survey, it has been included as a limitation of the study.

In the discussion, authors should mention that poor attitude towards LGBTQ+ individuals can affect resource utilizations and LGBTs health status (10.1016/j.msard.2017.02.001), also with implications for oral health.

Response: Thanks for the excellent suggestion, we included this information under discussion section of the manuscript.

Thank you again for your valuable time and effort in reviewing our manuscript.

I remain available for any questions.

Dr. Anubhuti Shukla

---

## [Editor Report · Decision Letter 1]

8 Feb 2022

Reviewing Challenges in Access to Oral Health Services Among the LGBTQ+ Community in Indiana and Michigan: A Cross-Sectional, Exploratory Study

PONE-D-21-36125R1

We’re pleased to inform you that your manuscript has been judged scientifically suitable for publication and will be formally accepted for publication once it meets all outstanding technical requirements.

Kind regards,

Luigi Lavorgna

Academic Editor

PLOS ONE
---

## [Editor Report · Acceptance letter]

10 Feb 2022

PONE-D-21-36125R1 

Reviewing challenges in access to oral health services among the LGBTQ+ community in Indiana and Michigan: A cross-sectional, exploratory study 

Dear Dr. Shukla:

I'm pleased to inform you that your manuscript has been deemed suitable for publication in PLOS ONE. Congratulations! Your manuscript is now with our production department. 

Kind regards, 

on behalf of

Dr. Luigi Lavorgna 

Academic Editor

PLOS ONE